# GO/TiO_2_-Related Nanocomposites as Photocatalysts for Pollutant Removal in Wastewater Treatment

**DOI:** 10.3390/nano12193536

**Published:** 2022-10-10

**Authors:** Ethan Dern Huang Kong, Jenny Hui Foong Chau, Chin Wei Lai, Cheng Seong Khe, Gaurav Sharma, Amit Kumar, Suchart Siengchin, Mavinkere Rangappa Sanjay

**Affiliations:** 1Nanotechnology and Catalysis Research Centre (NANOCAT), Institute for Advanced Studies (IAS), University of Malaya, Kuala Lumpur 50603, Malaysia; 2Department of Fundamental and Applied Sciences, Universiti Teknologi PETRONAS (UTP), Seri Iskandar 32610, Malaysia; 3Nanshan District Key Lab for Biopolymer and Safety Evaluation, Guangdong Research Center for Interfacial Engineering of Functional Materials, Shenzhen Key Laboratory of Polymer Science and Technology, College of Material Science and Engineering, Shenzhen University, Shenzhen 518060, China; 4International Research Center of Nanotechnology for Himalayan Sustainability (IRCNHS), Shoolini University, Solan 173229, India; 5School of Science and Technology, Global University, Saharanpur 247001, India; 6Natural Composites Research Group Lab, Department of Materials and Production Engineering, The Sirindhorn International Thai-German Graduate School of Engineering (TGGS), King Mongkut’s University of Technology North Bangkok (KMUTNB), Bangkok 10800, Thailand

**Keywords:** graphene oxide, titanium dioxide, photocatalysis, dye, heavy metals, oil

## Abstract

Water pollution has been a prevalent issue globally for some time. Some pollutants are released into the water system without treatment, making the water not suitable for consumption. This problem may lead to more grave problems in the future including the destruction of the ecosystem along with the organisms inhabiting it, and illness and diseases endangering human health. Conventional methods have been implemented to remove hazardous pollutants such as dyes, heavy metals, and oil but are incapable of doing so due to economic restraints and the inability to degrade the pollutants, leading to secondary pollution. Photocatalysis is a more recently applied concept and is proven to be able to completely remove and degrade pollutants into simpler organic compounds. Titanium dioxide (TiO_2_) is a fine example of a photocatalyst owing to its cost-effectiveness and superb efficiency. However, issues such as the high recombination rate of photogenerated electrons along with positive holes while being only limited to UV irradiation need to be addressed. Carbonaceous materials such as graphene oxide (GO) can overcome such issues by reducing the recombination rate and providing a platform for adsorption accompanied by photocatalytic degradation of TiO_2_. The history and development of the synthesis of GO will be discussed, followed by the methods used for GO/TiO_2_ synthesis. The hybrid of GO/TiO_2_ as a photocatalyst has received some attention in the application of wastewater treatment due to its efficiency and it being environmentally benign. This review paper thereby aims to identify the origins of different pollutants followed by the sickness they may potentially inflict. Recent findings, including that GO/TiO_2_-related nanocomposites can remove pollutants from the water system, and on the photodegradation mechanism for pollutants including aromatic dyes, heavy metal and crude oil, will be briefly discussed in this review. Moreover, several crucial factors that affect the performance of photocatalysis in pollutant removal will be discussed as well. Therefore, this paper presents a critical review of recent achievements in the use of GO/TiO_2_-related nanocomposites and photocatalysis for removing various pollutants in wastewater treatment.

## 1. Introduction

Water is an irreplaceable natural resource that has brought life to many civilizations and many lives depend on it. Thus, water holds the key to maintain the Earth’s ecology and quality of life. Of all available sources on the Earth, 97% of it is saline and the remaining 3% is drinkable water. UN World Water Development reported that currently 2.2 billion people do not have access to safe drinking water, and 4.2 billion, or 55% of the world’s population, are without safe sanitation [1]. As the years go by, the demand for clean water has increased due to increasing populations, water wastage, sudden climate change, insufficient rainfall, and mainly, water pollution, leading to water scarcity. Addressing water pollution has turned into a top priority globally. In this regard, solutions for effective wastewater treatment need to be introduced and implemented to mitigate the effects of pollutants on the ecosystem and human life that are highly dependent on water while meeting the increasing demand for clean water as supplies decrease due to pollution.

The pollutants chosen in this review are dyes, heavy metals, and oil. The presence of dyes with their aromatic and recalcitrant nature makes them hard to remove via conventional methods and a threat to human life. Heavy metals are also known for food poisoning and consumption of them has resulted in multiple terminal illnesses in human history. This review paper aims to discuss (i) the origin and effects of pollutants, (ii) GO and TiO_2_ as a photocatalyst, (iii) synthesis of GO and GO/TiO_2_, and (iv) recent findings on GO/TiO_2_-related nanocomposites in pollutant removal for wastewater treatment.

## 2. Dye

In modern times, dyes have been used extensively in many fields which require their products to be colored, ranging from wool to leather, textile, paper, rubber, and plastics [2,3,4]. Among the major industries that are known to contribute to the presence of dye effluents in the environment, the textile industry has the highest percentage. According to world trade export data from 2018, China is the major exporter of textiles, followed by the European Union and India [5], as shown in Figure 1. Without any proper treatment prior, dyes are released into the ecosystem. Due to their aromatic structure, they are non-biodegradable and may have a deleterious effect on both the ecosystem and the health of human beings dependent on the water source for their daily activities [6]. The presence of dyes on the surface and subsurface of water not only makes the environment less attractive but also may potentially be a source of water-borne diseases, viz. mucous membrane dermatitis, perforation of the nasal septum, and severe irritation of the respiratory tract [7]. At the same time, dyes obstruct the pathway of sunlight in the water, thus blocking the photosynthesis of aquatic plants. Most synthetic dyes are highly toxic, mutagenic, and carcinogenic, thus posing problems to the environment and human beings [8,9]. Several types of synthetic dyes can be categorized based on their molecular structure, for instance, azo dyes and anthraquinone dyes (see Table 1).

## 3. Heavy Metals

### 3.1. Arsenic

Arsenic is a metalloid and the 33rd element in the periodic table has the electron configuration of [Ar] 3d^10^ 4s^2^ 4p^3^ with the given notation of ‘As’ [11]. Arsenic is a ubiquitous element in the Earth’s crust and can be found in many different parts of the world [12]. Arsenic is introduced to water sources due to anthropogenic activities such as combustion of fossil fuels, use of pesticides and herbicides against pests, and mining and smelting, which involves metal ores produced other than by natural processes. Arsenic can exist in different forms with different charges, which are −3 (arsine), 0 (elemental arsenic), +3 (arsenite) and +5 (arsenate) [13]. As(V) commonly exist as oxyanions (H_2_AsO_4_^−^ or HAsO_4_^2−^) while As (III) species are protonated and form H_3_AsO_3_ which characterize them as an inorganic species of arsenic [14]. These inorganic species are known to be predominant in the environment as opposed to organic species. Thus, the ingestion of any of the inorganic arsenic compounds will have more severe effects on human health compared to organic arsenic compounds. Nevertheless, acute arsenic poisoning encompasses vomiting and diarrhea, but also death in extreme scenarios. Excess intake of arsenic can lead to a damaged nervous system, gastrointestinal system, and renal system. At the same time, arsenic poisoning potentially results in skin lesions, diabetes, cardiovascular diseases, and cancer [15].

### 3.2. Mercury

Mercury is an element with the atomic number of 80 that is located in the d-block in the periodic table. Mercury is a unique metal whereby it takes in the form of liquid at room temperature. Natural sources include regions with active volcanic activity along with geothermal activity; other than that, human activities such as manufacture of pharmaceuticals and mining can lead to the liberation of mercury into the environment [16,17]. A neurological toxin, MeHg, which is commonly discussed in studies of mercury, poses a threat to aquatic life as well as humans. This hazardous compound being present in water sources will gradually bioaccumulate in fishes which, when consumed by humans, can lead to nuanced clinical issues [18]. Adverse effects of mercury studied on the human body are neurotoxicity, immunotoxicity, damaged kidneys, and potential deformation of the embryo [19].

### 3.3. Lead

Lead is an element with atomic number 82 and is a heavy metal nonessential for humans. The chemical symbol for lead, which is Pb, is taken from the word plumbum in Latin [20]. There are three valence states of lead known so far, which are 0 (elemental lead), II, and IV, with II being the most commonly discussed. The sources of lead include paints, electronic applications, mining, smelting, and leaded gasoline [21,22,23]. Lead can have adverse effects on organs throughout the human body, including the nervous and reproductive systems, even hematopoietic and renal organs due to oxidative stress [24]. Undetected lead in the body can also cause learning difficulties, retarded growth, seizures, and coma [25].

### 3.4. Cadmium

Cadmium has an atomic number of 48 which is located directly above mercury in the periodic table. Cadmium is usually found in compounds by combining with other elements, for instance, oxygen (cadmium oxide), chlorine (cadmium chloride), and sulfur (cadmium sulfate, cadmium sulfide). Cadmium can also come from human activities such as mining, smelting, industrial emissions, and agrochemicals [26,27]. Itai-itai has been a disease people suffered due to chronic cadmium-polluted rice fields [28]. Cancer, diabetes mellitus, and osteoporosis are likely to occur upon exposure to cadmium [29,30]. 

### 3.5. Chromium

Chromium is the 24th element in the periodic table and has the chemical notation of [Ar]3d^5^4s^1^. Chromium is commonly used in metal plating and alloying, paint manufacturing, and wood processing [31,32]. Acute reactions may occur which include breathing difficulties, skin problems, cancer, etc. [33]. Table 2 summarizes some adverse effects of metal ions on human beings.

## 4. Oil

Oil has also been one of the contributors to water pollution to date. The catastrophic event at the Deepwater Horizon in 2010 has led to a series of studies on approaches to deal with such situations if they ever reoccur. Other than that, oil can come from sources such as accidental oil spills from the petroleum hydrocarbon industry, and wastewater from industry and household use [46,47]. The contamination of oil in water sources can severely affect our livelihood and also our surroundings [48,49]. Oil can potentially cause cancer and other health complications in the body [50]. At the same time, oil can also disturb the ecological balance by changes in habitat and landscape, thus making oil a pollutant to look into and study [51].

## 5. Conventional Wastewater Treatment Methods for Waste Removal

With the rapid development of industry in recent years, and the manufacturing industry in particular, the amount of untreated organic waste being released into the environment has gradually increased, and it also poses potential health problems to both aquatic life and humans. Many methods have been developed and implemented over the years to control pollution. Conventional wastewater treatment methods can be split into three main branches which are chemical, physical, and biological methods. Table 3 displays the strengths and weaknesses of different methods. 

Physical methods are methods that separate organic compounds from wastewater via physical means such as adsorption and membrane filtration, which are simple yet effective. Adsorption itself is simple and proved to be effective for dye removal. The raw materials for adsorbent are abundant and the adsorbent can be recycled and regenerated for any future removal if required. However, some adsorbents can be costly themselves [64]. Examples of common adsorbents used are chitosan [65] and activated carbon [66]. Nanofiltration processes are driven by pressure passing through a membrane. The merits of using nanofiltration are that it has a higher flux in comparison to reverse osmosis and a higher rejection rate as compared to ultrafiltration [67]. Nevertheless, nanofiltration membrane fouling has been a drawback the process brings [68].

Biological methods introduce the use of microorganisms to degrade and adsorb pollutants in wastewater. Common biological methods utilize plants, also known as phytoremediation and bioremediation by microorganisms. The processes microorganisms used to remove dyes are an aerobic process, an anaerobic process, or a combination of both. Biological methods are still used as they are inexpensive and applicable to many organic wastes [69]. The problem with this approach is the time required for the organisms to degrade, while some organic compounds such as dyes are resistant to aerobic treatment [70].

Finally, chemical methods use chemical reagents or chemically created radicals as main components to either oxidize or degrade recalcitrant pollutants. Examples of such methods include coagulation, flocculation, ozonation, ion exchange, and advanced oxidation processes. Coagulation involves adding coagulants into the medium to destabilize the particles, increasing the tendency for them to form agglomerations, followed by flocculation, which allows the formation of larger agglomerates [71,72]. For cases of wastewater with high intensity of dye molecules, the process may lead to excessive dependence on coagulant which later forms chemical sludge and is not able to effectively remove dyes with distinct chromophore structures [73]. Ozone (O_3_) has been used to oxidize specific dyes with amine groups and aromatic structures [74]. However, the oxidation of O_3_ has been selective, therefore it is not suitable for some dyes, and as a result, unable to completely degrade organic dyes and can even produce toxic intermediates [75]. Photocatalysis is one of the promising advanced oxidation process (AOP) methods used to degrade organic wastes into harmless end products of carbon dioxide (CO_2_), water molecules (H_2_O), and mineral acids.

Photocatalysis involves a semiconductor in which electrons are excited from the valence band to the conduction band upon irradiation by a light source with energy equivalent or greater to the bandgap of the semiconductor [76,77]. The implementation of photocatalysis can overcome the disadvantages of other organic waste removal methods, providing complete degradation of organic pollutants and reducing operation costs because the irradiation source can be from renewable sunlight. As a result, the ease of application has made it one of the commonly practiced methods in wastewater treatment [78]. 

When the photocatalyst is exposed to a light source, the photons will be absorbed by the electrons, causing them to vacate the valence band and travel to the conduction band, leaving a positively charged hole, thus forming an electron-hole pair. The holes at the valence band will oxidize the water molecules nearby to form hydrogen ions and hydroxyl radicals, which are the primary radicals for pollutant degradation [79]. At the same time, the electron at the conduction band will convert the surrounding oxygen molecules into superoxide radicals. Subsequently, these radicals will tackle the dye molecules and degrade them. The overall process is displayed in Figure 2. Examples of photocatalysts used in photocatalysis are GO and TiO_2_.

### 5.1. TiO_2_

Titanium (IV) oxide, or titania for short, has been a promising material for wastewater treatment on an industrial scale [80]. Due to its nontoxicity, chemical stability, low cost, and other advantages, TiO_2_ has been intensely studied for its potential applications in the decomposition of organic compounds via redox reactions [81,82].

TiO_2_ is an n-type semiconductor with a wide indirect bandgap. The TiO_2_ structure consists of chains of twisted octahedra of TiO6, where each atom of Ti is surrounded by six oxygen atoms. Table 4 shows the three-dimensional stacking of the octahedra in rutile, anatase, and brookite. The tetragonal anatase unit cell contains four units of TiO_2_ (12 atoms), while the tetragonal rutile unit cell contains two units of TiO_2_ (6 atoms), and the orthorhombic brookite unit cell contains eight units of TiO_2_ (24 atoms). Anatase thus has a lower number of cells than rutile and brookite.

Rutile is the most thermodynamically stable phase of the three crystalline phases, which can withstand all temperatures and pressure due to its lower free energy. The anatase and brookite phases are metastable and appear to transition at higher temperatures to the rutile phase. Various sequences, such as anatase to brookite to rutile, brookite to anatase to rutile, brookite to rutile, and anatase to rutile can be used for the phase transformation. These transformations depend on temperature, time, and particle size. Anatase and brookite, at small particle sizes, are very stable. Despite the various advantages of photocatalysts, they have some drawbacks such as that they only can be activated by ultraviolet light, which is only about 5% of the solar spectrum and involves fast recombination of electron-hole pairs, limiting photocatalytic efficiency [84]. Ways to improve the photocatalytic efficiency of photocatalysts have been studied via doping with metals along with nonmetals, and in combination with other semiconductors [85,86]. Herein, we discuss the synthesis methods of GO and GO/TiO_2_ composite photocatalysts.

### 5.2. GO

In recent years, carbonaceous materials became an interest in studies of photocatalysts due to their physicochemical properties. A well-known allotrope of graphene, GO is primarily a layer of carbon atoms with a combination of sp^2^ and sp^3^ bonds due to oxidation of graphite, which is a distinct difference from graphene, which possesses only sp^2^ hybridized carbon atoms. The sp^3^ bonds are mainly attributed to various oxygenated functional groups that can be found covalently bonded on the surface of GO, for example, epoxy, hydroxyl, ketone, and carboxyl groups [87,88]. The introduction of functional groups has disrupted the symmetry of the π-network between carbon atoms in the graphene lattice, which restricts GO in many field applications [89]. However, they allow GO to become hydrophilic and disperse well in water, creating stable dispersions, hence leading to interesting future prospects in nanocomposites [90,91]. Functionalized GO-based nanocomposites are common photocatalysts due to their large specific surface areas and remarkable adsorption capabilities, and provide opportunities for further surface modification [92]. The functional groups that are on the basal plane of graphene provide a large surface area, making GO an excellent adsorbent for the removal of metal ions and dyes from aqueous solutions [93,94].

At the same time, GO can be further reduced into rGO via thermal, microwave, photochemical, microbial/bacterial, and chemical processes, using different reducing agents such as hydrazine, hydrogen sulfide, sodium borohydride, etc. [95,96,97]. The sp^2^ to sp^3^ bonds can be modified, depending on the degree of oxidation, which will impact the properties of GO. The electrical conductivity of GO was revived due to the reduction in sp^3^ which restores some graphene-like properties. The optical properties of rGO therefore should be between those of GO and graphene as the bandgap is being reduced [98]. Sharing similarities with GO, rGO is also responsive to most of the light spectrum and able to disperse in solvents credited to the remaining functional groups after reduction [99].

### 5.3. Production of GO

There are two mainstream approaches to synthesize GO: bottom-up methods whereby simple carbon molecules are used to generate pristine graphene, and top-down methods whereby layers of graphite are used to extract graphene sheets [100]. The drawbacks of bottom-up techniques are that they are time-consuming and are not suitable for large-scale production yet. Bottom-up techniques include chemical vapor deposition, epitaxial growth on silicon carbide, and so on. Therefore, the top-down method would be a more favorable and preferred option in the production of graphene derivatives. The products that are first obtained are graphene oxide or even reduced graphene oxide, which are well established in the application of nanocomposite materials.

Graphene oxide synthesis was originally described by Brodie, followed by Staudenmaier along with Hummers and Offeman. They developed their methods of graphene oxide synthesis via oxidation of graphite with improvements compared to the prior methods. Brodie’s work in 1859 reported mixing a ratio of 1:3 of graphite and potassium chlorate (KClO_3_), followed by the addition of fuming nitric acid (HNO_3_) in 3 or 4 days at a temperature of 60 °C [101]. The problem with this method is that it is time-consuming, complex, and hazardous because toxic ClO_2_ gas is released. The process was later improved by Staudenmaier in 1898 with the addition of concentrated sulfuric acid and replacing two-thirds of the fuming nitric acid in Brodie’s method, resulting in a highly oxidized GO in a one-step process. Still, the Staudenmaier method has a high risk due to the addition of KClO_3_, which may lead to explosion and is also time-consuming.

In 1958, Hummers et al. introduced KMnO_4_ to oxidize graphite sheets instead of KClO_3_ as it is safer. At the same time, sodium nitrate was used instead of fuming nitric acid to eliminate the production of fog acid. The advantages of Hummer’s method are that the process takes a few hours instead of a few days and yields high-quality GO. The disadvantage of Hummer’s method is that it produces toxic gases such as NO_2_ and N_2_O_4_. Nevertheless, the hazards of HNO_3_ and KClO_3_ discouraged any further use and have inspired new pathways for a safer and efficient method of graphene oxide synthesis. Modifications were made to Hummer’s method and Marcano et al.’s improvements, with an increased amount of KMnO_4_, and replacement of NaNO_3_ by substituting the combination of H_2_SO_4_ and H_3_PO_4_ with a ratio of 9:1 [102]. With the improvements, the method shows an even higher degree of oxidation and graphene product yield with enhanced hydrophilic properties. Figure 3 summarizes the different GO synthesis methods.

### 5.4. Production of TiO_2_

#### 5.4.1. Electrophoretic Deposition

Several factors make this the preferred method: efficiency in coating and film fabrication, shorter deposition time, film deposition on non-uniform surfaces, cost effectiveness, tunable thickness of films, homogenous coatings, and simple equipment requirements [104]. The process is initiated by a DC voltage which activates the charged particles in a suspended solution for deposition onto a substrate. An electric field due to voltage applied to the electrodes interacts with the surface charge of the nanoparticles, leading to the particles migrating to the electrode of opposite charge and the deposition on the electrode, resulting in formation of a homogenous layer.

#### 5.4.2. Spray Pyrolysis

Synthesis techniques such as spray pyrolysis involve a heated substrate, atomizer and a precursor solution (TiCl_3_ or Ti_4,_ etc.) [105]. This procedure produces thin films by atomizing the solution into tiny droplets, which are then transferred to the heated substrate. Due to the ultrasonic spraying technique used to create the smaller droplets, the atomic cloud aerosol produces larger droplets, at the same time influencing the surface morphology of material produced. Spray pyrolysis is incredibly effective, economical, and requires basic equipment as well. The thin films produced from this method also possess high substrate coverage and potential and homogeneity of mass synthesis. However, the drawbacks of this method include poor quality of thin film, thermal disintegration, and vapor convection, which are factors to be considered for synthesis of TiO_2_. Temperature differences cause the vapors to be produced, which prevents the source from adhering to the substrate.

#### 5.4.3. Sol Gel Method

Generally speaking, the sol-gel process entails transforming a system from a liquid “sol” phase, which is mostly in colloidal form, into a solid “gel” phase. Metal organic compounds, such as metal alkoxide and inorganic metal salts, are common precursors in the synthesis of “sol.” A sol or colloidal suspension is created by a succession of hydrolysis and polymerization processes. A wet “gel” will form when a sol is cast into a mold. The gel can be further dried and heated to create solid products.

Advantages of the sol-gel method include high surface area of synthesized materials, highly pure, simple equipment and low temperature conditions, and facile options for a range of processes from fiber to powder and coating [106]. The sol-gel approach has certain drawbacks as well, including very expensive precursor costs, a lengthy processing time, significant shrinkage during processing, and the tendency for hard agglomerates.

#### 5.4.4. Sonochemical and Microwave-Assisted Methods

Efficient photoactive TiO_2_ nanoparticles can also be synthesized via the sonochemical method while using ultrasonic irradiation for the hydrolysis of titanium tetraisopropoxide (TTIP) in pure water or in an ethanol/water mixture. The concept of acoustic cavitation causes the formation, growth, and collapse of bubbles in the solution while temperature at about 5000 K and pressures at about 1000 atm are the result of cavitational collapse.

On the other hand, electromagnetic waves such as microwaves have frequencies from 0.3 to 300 GHz along with wavelengths between 1 mm and 1 m, and can be used to synthesize TiO_2_ nanoparticles via microwave-assisted methods. Microwave heating offers a faster reaction time, a higher reaction rate, more selectivity, and a higher yield as opposed to conventional heating methods. Microwave heating can also be placed under two categories: (i) pulsed microwave heating and continuous microwave heating.

Microwave irradiation comes with the advantages of rapid heat transfer and selective heating. At the same time, energy can be evenly distributed within the sample, with improved reproducibility, and controllable experimental parameters.

### 5.5. Production of GO-TiO_2_

#### 5.5.1. Hydrothermal Method

Hydrothermal synthesis indicates a high-temperature and pressure technique for growing crystals from an aqueous solution in an autoclave. Figure 4 illustrates the common procedure for GO/TiO_2_ nanocomposites via the hydrothermal method. The characteristic of water is a solvent with a low boiling point, which allows it to be used under high pressure. Solvents with a high boiling point, such as dimethyl sulfoxide (DMSO), can be expensive and pose potential hazards, thus making water a very attractive option. Fine crystals of the desired nanocomposites are created using increased temperature. Hydrothermal synthesis enables the composition and consistency of the nanocrystals produced to be controlled. The key drawbacks associated with this process, however, are the inability to control material crystal growth (in the autoclave) and the cost of the equipment [107]. The hydrothermal reaction can also be used to partially reduce GO to graphene.

#### 5.5.2. Solvothermal Method

Solvothermal synthesis, analogous to the hydrothermal method, involves a process for fabricating crystals from non-aqueous organics using an autoclave at high temperatures and pressure [109]. Compared to the hydrothermal method, the solvothermal method typically has a greater effect on the size, shape, distribution, and crystallinity of the prepared nanocomposites.

#### 5.5.3. Mechanical Mixing

The simplicity and manipulations of the conditions of the reaction have made this strategy gradually gain popularity. This process involves mixing pristine or functionalized TiO_2_ with GO dispersions, likely accompanied by sonication and stirring to increase the optimum interaction between the precursors of the nanocomposite.

## 6. Heterojunction of GO/TiO_2_ Photocatalyst

GO acts as one of the superior support materials for different semiconductors and metals. From Figure 5, the photoelectrons generated at the TiO_2_ conduction band are rapidly transferred to the graphene layer by the TiO_2_ nanoparticle-decorated GO, which promotes the rate of organic dye pollutant degradation. In addition, during photocatalytic reactions, the high surface area of GO provides more surface adsorption sites for contaminants to greatly promote surface photocatalytic reactions, thus improving the catalytic activity. To summarize, graphene oxide has contributed in three ways to improving the photocatalytic degradation of pollutants: (1) improvement of the surface area of TiO_2_ due to its interaction with the two-dimensional matt structure of GO; (2) improvement of the adsorption of aromatic pollutants due to their strong π-π interactions with the aromatic network of GO; and (3) reduction of the recombination rate between the positive holes and the photogenerated electrons due to the significant electronic conductivity of GO, which functions as an electron sink for photogenerated electrons on the TiO_2_ surface [110,111].

## 7. Advantages of GO/TiO_2_ Composite Structure

Research carried out by [113] showed that the surface area of GO/TiO_2_ composite (78.12 m^2^/g) is larger than that of bare TiO_2_ (57.01 m^2^/g) through the nitrogen adsorption/desorption measurement using the Brunauer–Emmett–Teller (BET) equation. The researchers showed that increase in the surface area enhanced the pollutant adsorption ability of GO/TiO_2_ composite to approximately 37% compared to that of the produced bare TiO_2_. The contact area between the produced photocatalyst and pollutants is enhanced [114]. Thus, the photodegradation performance of GO/TiO_2_ composite is better than that of bare TiO_2_. A large surface area provides more active sites for redox reaction [115] including separation and transfer of photogenerated electron-hole pairs [116], thus enhancing the photodegradation performance. Besides, large surface area also improves the utilization of light [116]. Jiang et al. [114] found GO/TiO_2_ composite provides a surface area of 307.34 m^2^/g, which is much better when compared to GO and TiO_2_, which have surface areas of 119.99 m^2^/g and 58.45 m^2^/g, respectively. Therefore, it can be concluded that the addition of GO can effortlessly increase the specific surface area of GO/TiO_2_ composite. The researchers proved that both the adsorption and photodegradation performances of the GO/TiO_2_ composite were improved compared to those of bare GO and TiO_2_. The photodegradation efficiency of the prepared GO/TiO_2_ composite reached 97.03% after 100 min, which is 5.43 times higher than that of bare TiO_2_ with 17.88%. Other research (by [117] demonstrated bare TiO_2_ had a surface area of 49 m^2^/g and GO/TiO_2_ composite a surface area of 92 m^2^/g.

Band gap absorption edge of bare TiO_2_ is 440 nm, which mostly adsorbs UV light to initiate the photodegradation process. GO has great absorption in the visible light range [114]. Thus, when TiO_2_ combines with GO, the absorption range of GO/TiO_2_ composite can be expanded up to a wavelength range of 800 nm [113]. Compared with GO and TiO_2_, the light absorption characteristics of GO/TiO_2_ composite are enhanced. The generated Ti^3+^ enhances the visible light absorption range [114]. Research by [113] showed that prepared bare TiO_2_ had a bandgap reading of 3.20 eV and the bandgap reading of GO/TiO_2_ composite was 2.80 eV using a Tauc plot. This outcome indicates GO/TiO_2_ composite can absorb visible light effectively and therefore, enhance the photodegradation performance under visible light irradiation. Hunge et al. [115] reported TiO_2_ with a bandgap reading of 3.11 eV and GO/TiO_2_ composite with a bandgap reading of 2.72 eV. The formation of titanium bonds with carbon in GO/TiO_2_ composite successfully shifts the bandgap of bare TiO_2_ from the ultraviolet to the visible region [118]. Interaction of GO with TiO_2_ produces intermediate states close to the valence band of TiO_2_ and this allows the adjustment of the forbidden band reading [118]. This also reduces the recombination rate of charge carriers and enhances the photodegradation performance. Experiments by Hernández-Majalca et al. [118] managed to shift the bandgap energy of bare TiO_2_ from 3.10 eV to 2.60 eV by producing a GO/TiO_2_ composite, so the produced photocatalyst was activated under visible light. The Ti-O-C and O-Ti-C bonds have the effect of impurities on TiO_2_ where they introduce intermediate states in the forbidden band of TiO_2_ which promote the photogeneration of electrons from photons of lower energy [114,118].

GO/TiO_2_ composite possess large surface area and a low photogenerated electron-hole pairs recombination rate to degrade the pollutant. Other than that, the visible light absorption range of the GO/TiO_2_ composite can also be enhanced due to the response of graphene to visible light [114]. Electrochemical impedance spectroscopy (EIS) and photocurrent study can be carried out using a three-electrode system consisting of FTO covered with a sample as a working electrode, a platinum plate as counter electrode, and Ag/AgCl as a reference electrode. Figure 6 displays the EIS and photocurrent plots. Jiang et al. [114] reported GO/TiO_2_ composite showed a smaller semicircle compared to both bare GO and TiO_2_, indicating the GO/TiO_2_ composite as having the smallest charge transfer resistance. The lower the charge transfer resistance, the higher the efficiency of the charge carrier transfer process. In the figure, the photocurrent plot demonstrates that the photocatalyst is highly sensitive to light and the photocurrent density of the prepared GO/TiO_2_ composite is higher when compared to GO, which can be due to the effective charge separation and lower recombination rate of photogenerated electron-hole pairs.

## 8. Recent Studies on GO-TiO_2-_Related Nanocomposites in Wastewater Treatment and Their Findings

The GO-TiO_2_ heterostructure is one of the most significant and frequently studied semiconductor photocatalysts for the decomposition of organic pollutants [119]. Factors such as high surface area, excellent conductivity, and chemical stability allow carbonaceous materials such as graphene oxide to be a compactible surface modification combination with titanium dioxide. Hence, a plethora of studies have investigated and reported on graphene oxide-coupled TiO_2_, including other semiconductor materials for enhanced photocatalytic performance over organic pollutant degradation. Lin et al. reported on synthesized reduced graphene oxide decorated with titanium dioxide nanocomposites via a combination of ultrasonication and hydrothermal reaction which involved titanium tetrachloride (TiCl_3_) and graphene oxide (GO) as precursors [120]. As a result of combining the two components, the adsorption and degradation of methyl orange (MO) were significantly increased. The prepared sample was able to remove up to almost 90% of the dye, which was nearly threefold the rates of bare TiO_2_ and GO under ultraviolet lamp irradiation for 4 h. The process having lower degradation efficiency in sunlight than using ultraviolet illumination is due to the excitation of electrons in the conduction band being made easier by the high energy of ultraviolet light. The merit of a hybrid allows a greater surface area which contributes to the adsorption of MO onto GO surfaces via π-π interaction, followed by degradation of MO by active groups generated by TiO_2_, feats that cannot be accomplished by bare TiO_2_ and GO alone.

Qi et al. [121] studied the addition of nanosized metallic Ag particles on graphene oxide-titanium dioxide mesocrystals. The ternary composite was synthesized via the photoreduction deposition method. The catalyst was then used to degrade Rhodamine B (RhB) under the irradiation of a 300 W Xe lamp as a visible light source. The highlight of this study was that the ternary nanocomposite was able to nearly degrade RhB completely in less than 180 min, while the pristine titanium dioxide mesocrystals and graphene oxide-titanium dioxide mesocrystals were only able to remove about 60%. The photocatalytic activity of the three mentioned photocatalysts on RhB affirms the versatility and potential of graphene oxide-titanium dioxide as a photocatalyst in treating various water pollutants when doped with a noble metal. Adly et al. [122] investigated the application of nanostructured graphene oxide/titanium dioxide composites for photocatalytic degradation of rhodamine B (RhB) and acid green 25 (AG-25) dyes. The nanocomposite was synthesized by the hydrothermal method with different calcination temperatures. Referring to the literature, the highest degradation rate was achieved using 10 wt% GO/TiO_2_ treated at 400 °C in comparison to pristine TiO_2_ and other synthesized samples. RhB was completely removed after 1.25 h, and 96% of AG-25 was removed within 3 h. The obtained result was accredited to the increase in GO content to optimum values which allowed the incident UV and/or visible light to reach the TiO_2_ nanoparticles, thus leading to an increase in photocatalytic activity. GO plays a few important roles in nanocomposites whereby it can interact with organic contaminants due to the presence of functional groups via adsorption on the surface and an electron scavenger which prevents the recombination of electron-hole pairs of TiO_2_ nanoparticles. The initial concentration of dye plays a part in the photodegradation process as high concentration of dye will inhibit the light from reaching the photocatalyst and vice versa. At the same time, the photocatalyst loading depicts the removal efficiency whereby the increase of photocatalyst will result in an increase of photons and dye molecules adsorbed.

Sharma et al. conducted study involving GO-TiO_2_ nanocomposites synthesized using green alga *Chlorella pyrenoidosa* [123]. The model contaminant for this study was crystal violet (CV). The study was conducted under visible light whereby the photocatalytic of TiO_2_ nanoparticles and GO-TiO_2_ nanocomposite in degrading CV was evaluated. The study showed that the binary nanocomposite proved to be superior due to the reduction in bandgap and increased dye adsorption. Pristine TiO_2_ nanoparticles were able to degrade only 43% of CV, while GO-TiO_2_ was able to degrade 63% under the same conditions. Based on the calculations made, the photocatalytic activity of GO-TiO_2_ is double in comparison to that of TiO_2_ nanoparticles. Table 5 summarizes some GO/TiO_2_-based photocatalysts for photodegradation of dye while Table 6 are studies on the removal of arsenic ions with TiO_2_-based photocatalysts. In a work presented by Babu et al., Fe_2_O_3_/TiO_2_ was synthesized via sol-gel using P123 as a structure-directing agent [124]. The nanocomposite was later added to the surface of reduced graphene oxide, which can further increase the available active sites for arsenic ion adsorption. The nanocomposite was characterized by FTIR, Raman, XRD, and other analyses. The removal efficiency of Fe_2_O_3_/TiO_2_-rGO was tested and later compared to that of bare mesoporous Fe_2_O_3_/TiO_2_. The findings showed the former having a higher maximum adsorption capacity which was 99.5mg/g for As (V) and 77.7mg/g for As (III) due to enhanced active adsorption sites on the interface of the nanocomposite.

Zhang et al. prepared GO/TiO_2_ nanocomposites via the modified Hummers method and hydrothermal method [141]. The sample was irradiated under a 500 W Hg lamp as an alternate UV source in a solution mixed with Cd^2+^ and Pb^2+^ ions. According to their study, 66.32% of Cd^2+^ and 88.96% of Pb^2+^ were able to be photoreduced with 0.6 g L^−1^ of GO/TiO_2_ after 120 min at pH 6. GO/TiO_2_ was also used for Cr (VI) removal in a study by Hu et al. (2018). It was reported that nearly all Cr (VI) in the solution was reduced to Cr (III) after being adsorbed onto the surface of GO/TiO_2_ and reduced by electrons generated by UV light. Reduced GO/TiO_2_ nanowires with self-doped TiO_2_ were used for adsorption and degradation of waste engine oil [142]. The study found the novel nanocomposite allowed an outstanding 98.6% COD removal due to the synergic effect between TiO_2_ and GO under 300 W Xe lamp after 5 h. Qian et al. synthesized a membrane made of sulfonated GO/TiO_2_/Ag for separation and photodegradation of oil and water emulsion [143]. In that study, emulsions of oil and water were passed through the membrane while under UV irradiation with the assistance of gravity. The membrane was proven to have an efficiency of 99.6% while separating and degrading the emulsion successfully.

## 9. GO/TiO_2_ Photocatalytic Mechanism

The application of GO/TiO_2_ for the removal of aromatic pollutants such as dyes (methylene blue and methyl orange) reflects the efficiency of GO/TiO_2_ nanocomposites. Equations (1)–(8) describe the route taken by the photogenerated electrons followed by the decomposition of aromatic pollutants by GO/TiO_2_. The proposed mechanism of formation of radicals for the photodegradation of pollutants is in agreement with [126]. Lin et al. [120] also stated that dye molecules were initially adsorbed onto the surface of GO/TiO_2_ under the influence of π-π bonds of a graphene sheet.

The irradiation source then initiates the photogeneration of electron-hole pairs on the surface of photocatalyst. The resultant hole will break apart water molecules into hydrogen ions and hydroxyl radicals (OH^•^). The resultant electrons in the valence band of TiO_2_ are transferred to the conduction band of GO to convert oxygen molecules into hydrogen peroxide and subsequently OH^•^. This whole process repeats as long as the irradiation source still remains active.
(1)$$\mathrm{GO}/{\mathrm{TiO}}_{2}\to h^{+}+e^{-}$$
(2)$$h^{+}+{\mathrm{OH}}^{-}\to {\mathrm{OH}}^{•}$$
(3)$$h^{+}+H_{2}O\to H^{+}+{\mathrm{OH}}^{•}$$
(4)$$e^{-}+O_{2}\to O^{•2-}$$
(5)$$2e^{-}+{\;O}_{2}+2H^{+}\to {\;H}_{2}O_{2}$$
(6)$$e^{-}+{\;H}_{2}O_{2}\to {\;\mathrm{OH}}^{•}+{\;\mathrm{OH}}^{-}$$
(7)$${\mathrm{OH}}^{•}/O^{•2-}+\mathrm{dye}\to {\mathrm{CO}}_{2}+H_{2}O+\mathrm{simpler}\;\mathrm{compounds}\;$$
(8)$${\mathrm{OH}}^{•}/O^{•2-}+\mathrm{crude}\;\mathrm{oil}\to {\mathrm{CO}}_{2}+H_{2}O+\mathrm{simpler}\;\mathrm{compounds}\;$$

Band edge positions of the photocatalysts forming the GO/TiO_2_ composite have been determined for better estimation of the photogenerated charge carrier separation mechanism. The conduction band (CB) and valance band (VB) potentials of the photocatalysts represented by $E_{CB}$ and $E_{VB}$ can be calculated using Equations (9) and (10) [144] as below:(9)$$E_{CB}+0.50\;\left(E_{g}\right)=X-E_{e}$$
(10)$$E_{CB}=E_{VB}-E_{g}$$
where $E_{g}$ represents bandgap energy which can be obtained through the Tauc plot, X represents absolute electronegativity, and $E_{e}$ represents energy of free electrons on the hydrogen scale which has an approximate reading of 4.50 eV.

On the other hand, Mott-Schottky (MS) can be utilized to determine the rationality of the $E_{CB}$ and $E_{VB}$ readings calculated through the previous equations. By using the MS plot, we are able to determine the type of semiconductor or metal oxide. Based on the literature, GO is a p-type semiconductor whereas TiO_2_ is an n-type semiconductor [145]. Figure 7 displays the MS plot obtained through research by [145] showing p-type GO (negative slope of MS plot) and n-type TiO_2_ (positive slope of MS plot). The flat band potential reading can be determined through the MS plot by extending the MS plot to 1/C^2^ = 0 [144]. A study by [145] showed the flat-band potential reading of p-type GO and n-type TiO_2_ are determined to be 1.08 V and − 0.39 V (vs. Ag/AgCl, pH 6.50), respectively. The flat band potential can be converted to normal hydrogen scale using Equation (11) [145] below:(11)$$V_{\left(NHE,\;\;pH7\right)}=V_{\left(Ag/AgCl,pH6.5\right)}+0.21\;V-0.059\times \left(7-6.50\right)$$

After the conversion using Equation (11), the flat-band potential readings of the p-type GO and n-type TiO_2_ were 1.26 V and −0.21 V (vs. NHE, pH 7), respectively. The flat-band potential for the p-type semiconductor is 0.10 to 0.30 V lower than the reading of $E_{VB}$ whereas the flat-band potential for the n-type semiconductor is 0.10 to 0.30 V higher than the reading of $E_{CB}$ [144]. Therefore, the $E_{VB}$ of p-type GO was 1.56 V, while the $E_{CB}$ of n-type TiO_2_ was −0.51 V (vs. NHE, pH 7). Using Equation (9), the $E_{CB}$ of GO and $E_{VB}$ of TiO_2_ were determined to be −1.94 V and 2.59 V, respectively. After confirming both the $E_{CB}$ and $E_{VB}$ readings, the overall photocatalysis mechanism of the GO/TiO_2_ composite can be estimated. Figure 8 demonstrates the mechanism of GO/TiO_2_ composite.

The removal of heavy metals by GO/TiO_2_ nanocomposites is commonly via adsorption or photocatalytic reduction. The removal of Cr (VI) using GO/TiO_2_ was studied by [146] with the aid of XPS analysis, as described in Figure 9. The negatively charged Cr(VI) ions are attracted to the positively charged surface of the GO/TiO_2_ nanocomposite (step a). The photogenerated electrons then reduce the Cr(VI) ions into Cr(III) ions (step b). Lastly, the positively charged Cr(III) ions are captured by the functional groups of GO/TiO_2_ (step c).

## 10. Factors Affecting the Degradation of Organic Pollutants

### 10.1. Nature of Photocatalyst

Due to variations in the lattice structure, morphology, surface area, particle size, as well as impurities on the catalyst surface, different photocatalysts will have different photocatalytic activities, thus, influencing the adsorption of pollutants at the surface of the photocatalyst, and the rate of recombination of electron-hole pairs. For example, the three phases of TiO_2_ have different lattice structures in nature, which result in a slight difference in energy bandgaps. However, the efficiency of anatase outshines that of the others due to very stable surface peroxide groups being able to form in anatase TiO_2_. In recent studies, it was reported that the combination of anatase and rutile has improved performance compared to that of bare of anatase and rutile. Other than that, the surface area is also directly related to the active sites for the reaction, thus the degradation efficiency. The higher surface area contributes to the greater number of active sites leading to more organic molecules being adsorbed for degradation, improving the degradation efficiency.

### 10.2. Effect of pH

One of the important variables that will affect the performance of dye degradation is pH as it influences the dye reactions in several ways. A study conducted by [147] mentioned that titanium dioxide tends to have better photocatalytic activity in acidic mediums, with pH 5 being the most suitable for phenol degradation [147]. The zero-point charge of TiO_2_ is 6.2; under such conditions, the surface of TiO_2_ becomes positively charged. At the same time, the pH of a solution can also affect the photodegradation rate of different dyes due to their respective charges and interactions with the photocatalyst surface. GO-TiO_2_ is effective in synthetic wastewater which contains methylene blue with a pH value between 5.94 and 7.98 and the inverse when pH is at 1.90 [148]. The GO surface becomes negatively charged under basic solutions due to the deprotonation of oxygenated functional groups [149]. Sheshmani and Nayebi reported that the highest Remazol Black B removal was at pH 4 with an efficiency of 36.5% [150]. The addition of GO made the TiO_2_ surface more positively charged. In acidic and neutral conditions, the opposing charges of the photocatalyst and dyes promote adsorption, leading to effective dye removal.

### 10.3. Initial Pollutant Concentration

Adsorption of pollutants onto the surface of the photocatalyst can contribute to the efficiency of the photocatalyst in the removal of pollutants. Pollutant adsorption depends on the initial pollutant concentration in the solution. The initial concentrations of the pollutant can thus influence the process of photodegradation. The pollutants that are adsorbed on the surface and not the bulk of the solution are involved in the process. High concentrations of pollutants will increase the turbidity of water and cover more active sites. Consequently, fewer photons will reach the surface; fewer OH_2_ species will therefore be produced, leading to a decrease in the efficiency of degradation.

### 10.4. Photocatalyst Concentration

The optimum amount of photocatalyst needed usually depends on the concentration of the pollutant and the volume of the solution in which the pollutants are mixed. As the concentration of photocatalyst is below the optimum level, the rate of photocatalysis will gradually increase along with the increase in photocatalyst loading. This can be explained due to the increase in available active sites on the photocatalyst surface, and the number of radicals generated from the photocatalyst to degrade and mineralize the pollutant. However, when the photocatalyst is further added beyond the optimum level, the rate of photodegradation gradually decreases. At high concentrations, the photocatalyst will likely agglomerate and thus lead to unfavorable scattering of light and fewer photons can reach the surface of the photocatalyst [151].

## 11. Conclusions and Future Perspectives

Photocatalysis has proven to be an efficient method in pollutant removal for wastewater treatment with GO-TiO_2_ having a promising future. Factors that influence the rate of photodegradation of pollutants such as the nature of the photocatalysis, pH, the concentration of pollutants, and photocatalysts were briefly discussed. GO, a carbonaceous material that originates from graphene while being equipped with various functional groups, has allowed improvements in using TiO_2_ as an overall photocatalyst.

Despite the advanced developments and findings in photocatalysts, the applications of GO-TiO_2_ for industrial-scale wastewater treatment have been far and few in between. Plausible issues with the actual implementation as a photocatalyst could be related to the potential agglomeration of GO-TiO_2_ which limits industrial use. The agglomeration of GO-TiO_2_ reduces the overall surface area and results in a reduced rate of photodegradation. Therefore, more studies should be conducted to reduce the drawbacks of photocatalysis and further improve its performance in pollutant removal while reducing the cost.

## Figures and Tables

**Figure 1 nanomaterials-12-03536-f001:**
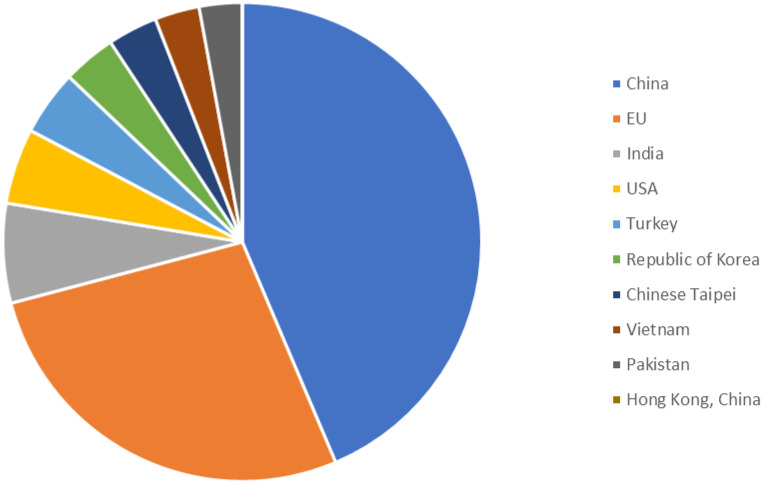
Major textile exporters in the world in 2018 [10].

**Figure 2 nanomaterials-12-03536-f002:**
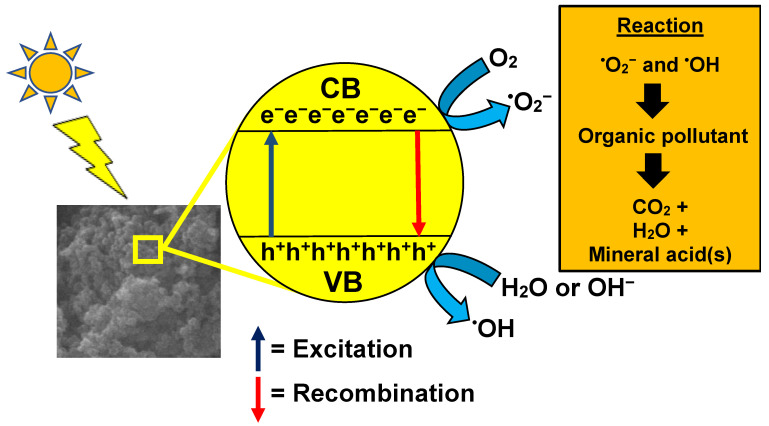
Mechanism of photocatalysis for organic waste removal.

**Figure 3 nanomaterials-12-03536-f003:**
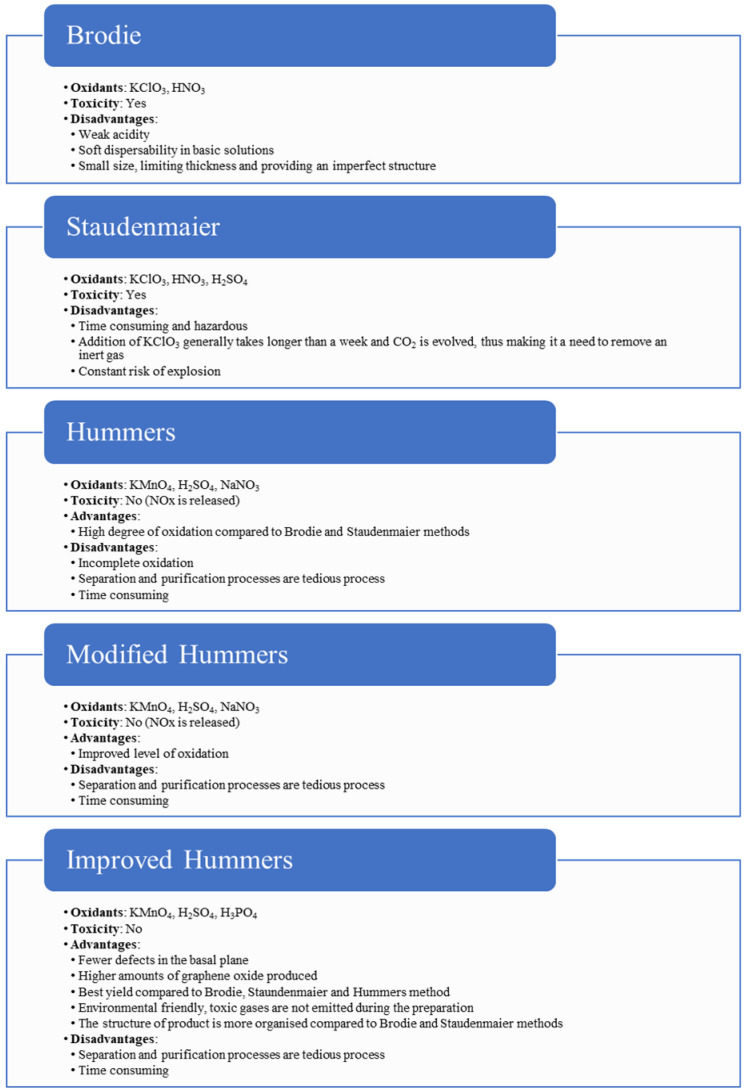
Synthesis methods of graphene oxide. Reprinted with permission from [103]. Copyright 2016 American Chemical Society.

**Figure 4 nanomaterials-12-03536-f004:**
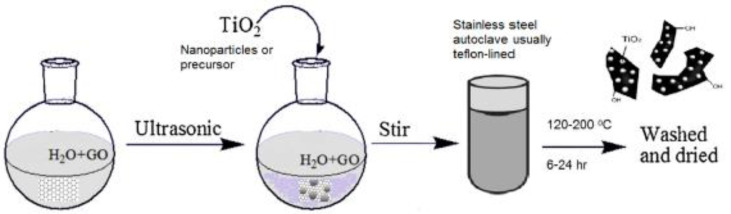
Schematic diagram of hydrothermal synthesis of GO-TiO_2_ nanocomposites. Reprinted with permission from [108] Copyright 2012 American Chemical Society.

**Figure 5 nanomaterials-12-03536-f005:**
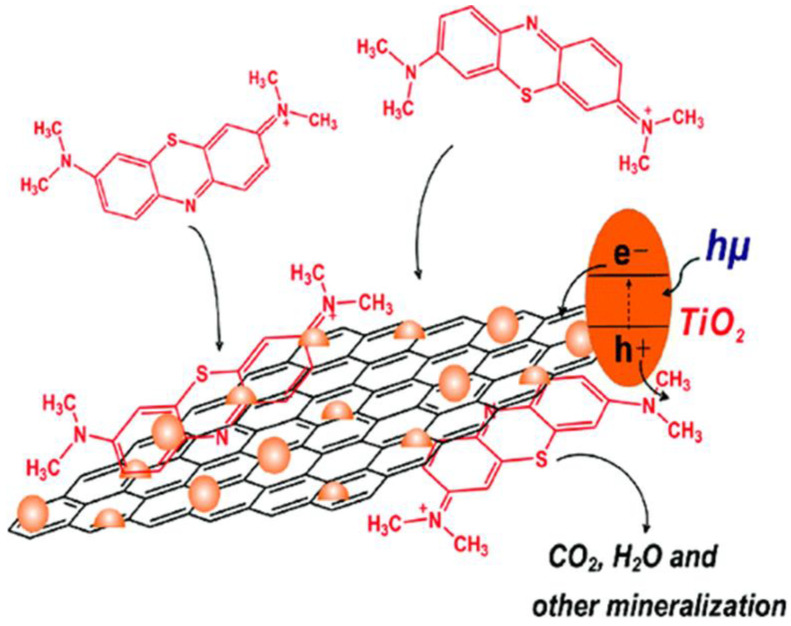
Photocatalysis mechanism of GO/TiO_2_ heterojunction. Reprinted with permission from [112]. Copyright 2010 American Chemical Society.

**Figure 6 nanomaterials-12-03536-f006:**
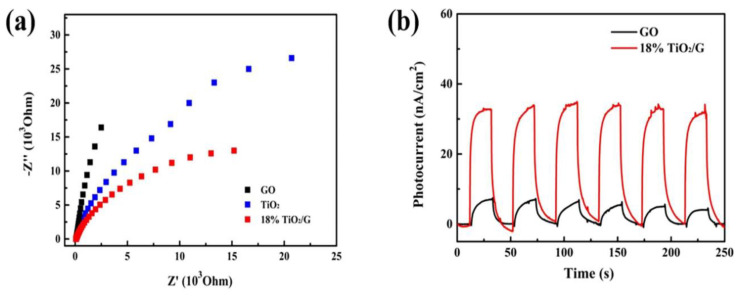
(**a**) EIS plot and (**b**) photocurrent plot. Reprinted with permission from [114]. Copyright Elsevier.

**Figure 7 nanomaterials-12-03536-f007:**
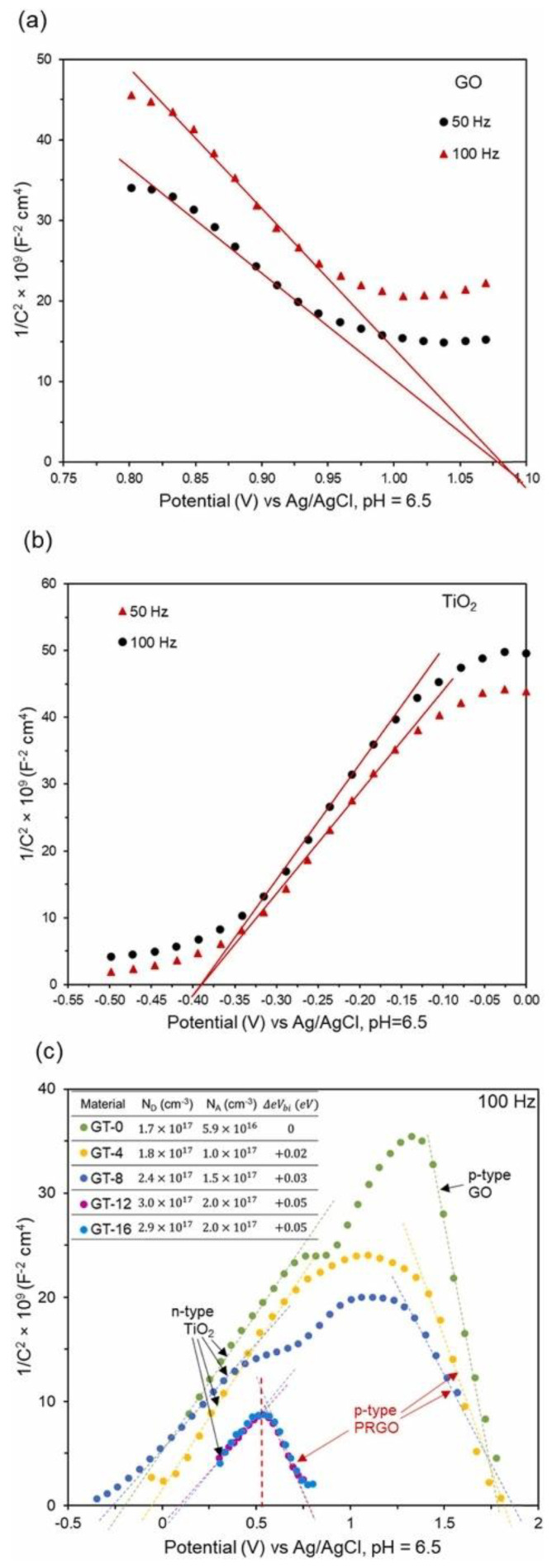
(**a**) MS plot of GO, (**b**) MS plot of TiO_2_, and (**c**) MS plot of GO/TiO_2_ composite. Reprinted with permission from [145]. Copyright Elsevier.

**Figure 8 nanomaterials-12-03536-f008:**
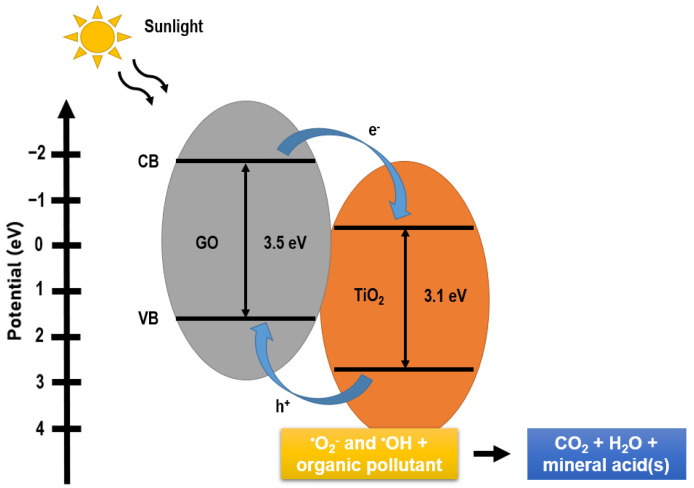
Mechanism of GO/TiO_2_ composite.

**Figure 9 nanomaterials-12-03536-f009:**
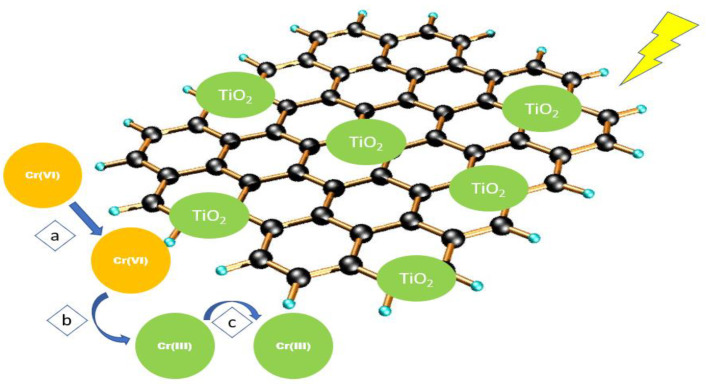
Removal mechanism of Cr(VI) using GO/TiO_2_ nanocomposite.

**Table 1 nanomaterials-12-03536-t001:** Chemical structures of dyes.

Dye	Chemical Structures
Methyl Orange	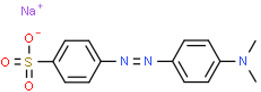
Congo Red	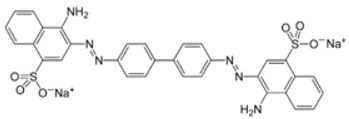
Direct Blue 1	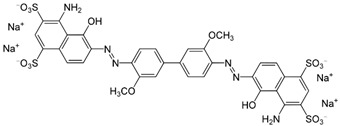
Remazol Brilliant Blue R	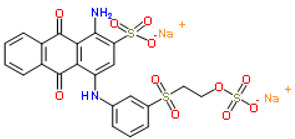
Alizarin	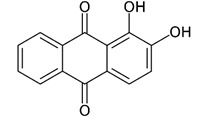

**Table 2 nanomaterials-12-03536-t002:** Adverse effects of different metal ions on human health.

Metal Ion	Adverse Effects on Human Health	Ref.
Hg^+^	Neurological alterations, motor dysfunction, and premature death	[18,34,35]
Pb^2+^	Impairment of brain and nervous functions, reproductive system, and miscarriage	[36,37,38]
Cd^2+^	Hypertension, teratogenic towards liver, kidney, and lungs	[39,40]
As^3+^	Skin lesions, diabetes, and cancers (e.g., skin, lung, kidney, bladder)	[41,42,43]
Cr^6+^	Damage hearing, skin problems, cancer	[31,44,45]

**Table 3 nanomaterials-12-03536-t003:** Strengths and weaknesses of various methods for organic waste removal.

Method	Strength	Weakness	Ref.
Coagulation–flocculation	Simple Economically friendly	Handling and disposal problems due to high sludge production High cost of chemical reagent High operational cost	[52]
Ozonation	Effective decolorization	High operational cost	[53]
Ion exchange	No loss of sorbents	Economically unattractive Not applicable to certain organic wastes	[54]
Biological treatment	Ecofriendly Energy saving	Time-consuming Occupy a certain area of land	[55]
Adsorption	High removal efficiency Low-cost Simple	Some adsorbents can be costly	[56,57]
Nanofiltration	Efficient Low energy consumption	Membrane fouling Membrane pore size restricted to nanopore size	[58,59,60]
Photocatalysis	Complete degradation of organic pollutant Production of harmless end products Usage of renewable sunlight energy Stable Inexpensive	Photocatalysts need to be activated by UV light Fast recombination of charge carrier	[61,62,63]

**Table 4 nanomaterials-12-03536-t004:** Properties of rutile, anatase, and brookite of TiO_2_ [83].

Phases	Crystal Structure	Band Gap (eV)	Space Group	Density (g/cm^3^)	Refractive Index	Structure Geometry
Rutile	Tetragonal	3.05	P4_2_/mnm	4.25	2.609	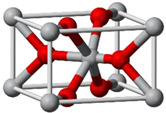
Anatase	Tetragonal	3.23	I4_1_/amd	3.894	2.488	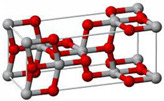
Brookite	Orthorhombic	3.26	Pbca	4.12	2.583	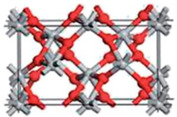

**Table 5 nanomaterials-12-03536-t005:** GO/TiO_2_-based photocatalyst for photodegradation of dye.

Photocatalyst	Light Source/Pollutants	Experimental Conditions	Photodegradation Efficiency (%)	Ref.
TiO_2_/GO/Ag	Solar irradiation Methyl Orange (MO)	Catalyst = 0.1 g [MO] = 15 mg/L Irradiation time = 180 min	~98	[86]
Ag/GO-TMCs	300 W Xe lamp Rhodamine B (RhB)	Catalyst = 0.5 g/L [RhB] = 20 mg/L Irradiation time = 180 min	~100	[121]
GO/TiO_2_	UV lamp Rhodamine B (RhB) and Acid Green 25 (AG-25)	Catalyst = 1.0 g/L [RhB] = 10 mg/L [AG-25] = 40 mg/L Irradiation time = 180 min	RhB = 100 AG-25 = 96	[122]
BiVO_4_/TiO_2_/GO	1000 W Xe lamp C.I. Reactive Blue 19 (RB-19)	Catalyst = 6 g/L [RB-19] = 0.05 mg/L Irradiation time = 90 min	95.87	[125]
GO/TiO_2_	40 W UV lamp Methylene Blue (MB) and Methyl Orange (MO)	Catalyst = 0.4 g/L Irradiation time = 25 min (MB) and 240 min (MO)	MB = 100 MO = 84	[126]
TiO_2_/Diazonium-GO	75 V filament lamps Methylene Blue (MB)	Catalyst = 0.2 g/L Irradiation time = 420 min	95	[127]
TiO_2_-Pt/GO	15W UV lamp Natural sunlight Amaranth Sunset yellow Tartrazine	Catalyst = 0.2 g/L [Amaranth] = 2 × 10^−5^ M [Sunset yellow] = 2 × 10^−5^ M [Tartrazine] = 2 × 10^−5^ M	Amaranth = 99.56 Sunset yellow = 99.15 Tartrazine = 96.23	[128]
TiO_2_/CaIn_2_S_4_@rGO	Visible light Methylene Blue (MB) and Congo Red (CR)	Catalyst = 1 mg [MB] = 35 mg/L Irradiation time = 30 min	~100	[129]
N-TiO_2_/Ag_3_PO_4_@GO	Visible light Acid Blue 25 dye (AB25)	Catalyst = 1 g/L [AB25] = 18 µM Irradiation time = 20 min	98	[130]
Cr_2_S_3_-GO/TiO_2_	500 W Xe lamp Methyl Blue (MB), Rhodamine B (RhB), and Methyl Orange (MO)	Catalyst = 0.4 g/L [MB] = 10 mg/L [RhB] = 10 mg/L [MO] = 10 mg/L Irradiation time = 120 min	~98	[131]
TiO_2_@rGO	Rhodamine-B dye (RhB)	Catalyst = 0.3 g/L [RhB] = 30 mg/L Irradiation time = 60 min	97	[132]
GO-TiO_2_	125 W medium pressure mercury lamps Acid Navy Blue dye (ANB)	Catalyst = 0.3 g/L [ANB] = 30 mg/L Irradiation time = 90 min	95	[133]
TiO_2_-RGO	150 W Xe lamp Rhodamine-B dye (RhB)	Catalyst = 0.4 g/L [RhB] = 0.001 mg/L Irradiation time = 180 min	~85	[134]
TiO_2_/GO	450 W lamp Methylene Blue (MB)	Catalyst = 1 g/L [MB] = 0.01 mM Irradiation time = 60 min	~51	[135]
TiO_2_/Fe_3_O_4_/GO	400 W UV lamp Methylene Blue (MB)	Catalyst = 0.1 g/L [MB] = 10mg/L Irradiation time = 90 min	~82	[136]
rGO-ZnS-TiO_2_	Crystal Violet dye (CV)	Catalyst = 0.4 g/L [CV] = 500 ppm Irradiation time = 50 min	~ 97	[137]

**Table 6 nanomaterials-12-03536-t006:** TiO_2_-based photocatalyst for heavy metal removal.

Photocatalyst	Light Source/Pollutants	Experimental Conditions	Photodegradation Efficiency (%)	Ref.
rGO-TiO_2_@fibers	As(V)	Catalyst = 9.3 mg/40 mL [As(V)] = 1 mg/L	97.0	[138]
Biochar TiO_2_	As(V)	Catalyst = 1 g/L [As(V)] = 50–300 mg/L	118.1	[139]
Hydrous TiO_2_	As(V)	Catalyst = 0.5 g/L [As(V)] = 20 mg/L	44.0	[140]

## Data Availability

All data generated or analyzed during this review are included in this manuscript.

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
