# Peer review of "GO/TiO_2_-Related Nanocomposites as Photocatalysts for Pollutant Removal in Wastewater Treatment"

_nanomaterials, 2022, doi:10.3390/nano12193536_

Round 1
Reviewer 1 Report
The manuscript is a review on GO-TiO2 photocatalyst for water purification from various organic and inorganic pollutants. In principle, the review is quite high-quality and shows modern views on the problem under discussion. I have only a few minor comments.
1. Table 5. Very few examples given. It should be expanded.
2. A new table should be given describing the removal of heavy metal and arsenic contamination from water by photocatalysis based on titanium dioxide.
3. The authors describe only titanium dioxide–GO composite; it is necessary to make at least a brief comparison of this system with other types of photocatalysts.
Author Response
Point 1: Table 5. Very few examples given. It should be expanded.
Response 1: Thank you for the suggestion. More examples have been added to Table 5.
Point 2: A new table should be given describing the removal of heavy metal and arsenic contamination from water by photocatalysis based on titanium dioxide.
Response 2: Thank you for the suggestion. Table 6 had been added with TiO2 based photocatalyst for heavy metal removal.
Point 3: The authors describe only titanium dioxide–GO composite; it is necessary to make at least a brief comparison of this system with other types of photocatalysts.
Response 3: Thank you for the suggestion. However, we would like to seek clarification regarding the types of photocatalyst the reviewer had in mind.

Reviewer 2 Report
Dear Editor: I would like to express my deep thanks for inviting me to review the manuscript ID: nanomaterials-1933726-peer-review-v1
Title: GO/TiO2 related nanocomposites as photocatalyst for various pollutants removal in wastewater treatment
Authors: Ethan Dern Huang Kong, Jenny Hui Foong Chau1, Chin Wei La*, Cheng Seong Khe, Gaurav Sharma, Amit Kumar, Suchart Siengchin and Mavinkere Rangappa Sanjay
Comments:
Authors briefly explained in the review articles on the GO/TiO2 nanocomposites and its photocatalyst wastewater treatment. However, there is no sufficient information for instance the synthesis process and their structure. Further, there is no sufficient information regards TiO2 structure related information and their photocatalyst activities. GO production is not relevant to this review. Therefore, it is suggested to rewrite the articles according to the relevant application and provide TiO2 nanocomposite related information instead of GO only.
There is also no sufficient information related to the photocatalytic activities results.
RECOMMENDATION
After reviewing the enclosed manuscript for “Nanomaterials”, the present manuscript contains some kinds of scientific analysis but it is mandatory required to modify according to the preceding remarks. So, the manuscript can be publication after major revision.
Author Response
Point 1: However, there is no sufficient information for instance the synthesis process and their structure.
Response 1: Thank you for the comment. We had added synthesis methods for the production of TiO2
Point 2: Further, there is no sufficient information regards TiO2 structure related information and their photocatalyst activities.
Response 2: Thank you for the comment. However, we have problems understanding what structure is the reviewer referring to and would like to seek further clarification on the problem raised.
Point 3: There is also no sufficient information related to the photocatalytic activities results.
Response 3: Thank you for the comment. We had added a section which explained the advantages of GO/TiO2 composite along with the performance of various photocatalysts.
Round 2
Reviewer 2 Report
Author addressed all the review comments in the revised review manuscript.